# Delay in healthcare seeking for young children with severe pneumonia at Mulago National Referral Hospital, Uganda: A mixed methods cross-sectional study

**Phiona Ekyaruhanga**[1,2]*, **Rebecca Nantanda**[2], **Hellen T. Aanyu**[3], **John Mukisa**[4], **Judith Amutuhaire Ssemasaazi**[5], **Mukeere John**[1], **Palma Aceng**[1], **Joseph Rujumba**[1]

1 Department of Paediatrics and Child Health, College of Health Sciences, Makerere University, Kampala, Uganda, 2 Makerere University Lung Institute, College of Health Sciences, Makerere University, Kampala, Uganda, 3 Department of Paediatrics and Child Health, Mulago National Referral Hospital, Kampala, Uganda, 4 Department of Immunology and Molecular Biology, College of Health Sciences, Makerere University, Kampala, Uganda, 5 Department of Clinical Epidemiology, College of Health Sciences, Makerere University, Kampala, Uganda

* ephiona@yahoo.com

**Data Availability Statement:** This minimal dataset for replicating findings from this study can be

## Abstract

### Background

Globally, pneumonia is the leading infectious cause of under-five mortality, and this can be reduced by prompt healthcare seeking. Data on factors associated with delays in seeking care for children with pneumonia in Uganda is scarce.

### Objectives

The study aimed to determine the prevalence, factors associated with delay, barriers, and facilitators of prompt healthcare seeking for children under five years of age with severe pneumonia attending Mulago National Referral Hospital (MNRH) Uganda.

### Methods

A mixed methods cross-sectional study was conducted among 384 caregivers of children with severe pneumonia at MNRH. Quantitative data was collected using interviewer-administered structured questionnaires and qualitative data through focus group discussions with caregivers. Descriptive statistics were used to determine the prevalence of delay in care seeking. Logistic regression analysis was used to determine the factors that were independently associated with delay in seeking healthcare. Content thematic analysis was used to analyze for barriers and facilitators of prompt healthcare seeking.

### Results

The prevalence of delay in seeking healthcare was 53.6% (95% CI: 48.6–58.6). Long distance to a hospital (AOR = 1.94, 95% CI 1.22–3.01, p value = 0.003), first seeking care elsewhere (AOR = 3.33, 95% CI 1.85–6.01, p value = 0.001), and monthly income $\leq 100,000$

found at https://github.com/Ekyaruhanga/Dataset_-Delay.

**Funding:** The authors received no specific funding for this work.

**Competing interests:** The authors have declared that no competing interests exist.

UGX (28 USD) (AOR = 2.27,95% CI 1.33–3.86, p value = 0.003) were independently associated with delay in seeking healthcare. Limited knowledge of symptoms, delayed referrals, self-medication, and low level of education were barriers to prompt healthcare seeking while recognition of symptoms of severe illness in the child, support from spouses, and availability of money for transport were key facilitators of early healthcare seeking.

## Conclusion

This study showed that more than half of the caregivers delayed seeking healthcare for their children with pneumonia symptoms. Caregivers who first sought care elsewhere, lived more than 5 km from the hospital, and earned less than 28 USD per month were more likely to delay seeking healthcare for their children with severe pneumonia. Limited knowledge of symptoms of pneumonia, self-medication, and delayed referral hindered prompt care-seeking. Key facilitators of prompt care-seeking were accessibility to health workers, support from spouses, and recognition of symptoms of severe illness in children. There is a need for programs that educate caregivers about pneumonia symptoms, in children less than five years.

## Introduction

Pneumonia is the leading infectious cause of death among children less than five years globally [1–4] with an estimated two children dying every minute. In 2020, global estimates indicated that over 800,000 children less than five years died of pneumonia [5]. According to the World Health Organization (WHO), the African region has the highest number, accounting for more than half of all deaths from childhood pneumonia [6]. In Uganda, pneumonia is the leading cause of child morbidity and mortality [7, 8]. Studies have indicated high pneumonia case fatality rates ranging between 10%-30% [9–11]. In 2018, a total of 95,413 admissions in the country were due to pneumonia, representing 13.4% of overall admissions in this age group. The outcome in 1,508 (9.5%) of the admissions was death [12].

Seeking healthcare within 24 hours of developing pneumonia symptoms is key to the prevention of complications and deaths in children less than five years [13]. However, this remains a big challenge in healthcare settings in low-income countries, thus contributing to preventable pneumonia-associated deaths [11]. This is despite the WHO-led strategies like the Integrated Community Case Management (iCCM) currently implemented in many parts of Uganda [14]. Delay, defined as seeking healthcare from a hospital or health center, after 24 hours of the onset of the pneumonia symptoms, may lead to worsening symptoms, severe complications like loss of consciousness, hypoxemia, convulsions, and in worst cases death [11]. The "three delays" model that was originally proposed to describe the determinants of maternal mortality [15] has been previously applied to understand delays in seeking care among caregivers of children with pneumonia in the Peruvian Amazons [16]. The model describes the first delay as a delay in deciding to seek appropriate care, the second as a delay in identifying and reaching the health facility, and the third as a delay in receiving healthcare once in the health facility. The model suggests that individuals who experience any of the delays may miss prompt receipt of appropriate care and are at high risk of severe complications like hypoxemia, loss of consciousness, and death [15, 16].

Studies in Ethiopia, Kenya, and Bangladesh showed delays in care-seeking for children with pneumonia in 48.6%, 62.1%, and 52.7% of cases respectively [17–19]. In these studies, the factors that were associated with delay in seeking care included; the cost of services or treatment, transportation costs and loss of wages incurred while taking the child to a hospital, sex of caregiver and social norms, insufficient knowledge of danger signs and illness severity, previous experiences with health services, child's age, and sex, maternal education, maternal employment status, decision-making status, use of self-prescribed antibiotics, gender roles, rural residence, perceived quality of health services, cultural beliefs about the illness and perceived illness severity [11, 18–21]. In Eastern Uganda, a case-series study found that the majority of children with severe forms of pneumonia often reported to treatment centers after 3 days from the onset of symptoms [11]. This study aimed to determine the prevalence of delay in seeking care for young children with severe pneumonia and explore the associated factors, in a tertiary care hospital in Uganda.

## Materials and methods

### Study design and setting

This was a mixed-methods cross-sectional study among 384 caregivers of children less than five years with severe pneumonia presenting at the paediatric emergency unit of Mulago Hospital, Kampala, between December 2020 and June 2021. Mulago Hospital is a National Referral Hospital in Uganda and a teaching hospital for the Makerere University College of Health Sciences, Kampala. It is the largest public hospital in Uganda with 1,500 beds [22]. Although it is a national referral, many patients especially those from Kampala city and surrounding districts use it as their first point of formal care. The paediatric emergency unit is a 50-bed capacity unit comprised of the paediatric intensive care unit (PICU) and, high and low-dependency wards. Children aged 1 day up to 12 years of age are usually admitted for 24 hours, given immediate treatment, and transferred to other paediatric wards for continued management. Paediatricians, paediatric residents, medical officers, intern doctors, and nurses provide care for this unit. Parents or caregivers of sick children seek care at the paediatric emergency unit either as referrals or as the first contact with the formal health system. An estimated 109 of 160 (68%) children with severe pneumonia seen at the paediatric emergency unit every month use it as the first point of formal healthcare.

### Sample size estimation

The sample size to determine the prevalence of delay in seeking healthcare was estimated using the formula for single proportions. We assumed a 95% level of confidence, 5% sampling error, proportion of caregivers that delayed seeking healthcare for their children of 47.7% [11]. This gave an estimated sample size of 384. In addition, we estimated the sample size for determining factors associated with delay in seeking healthcare based on the formula for comparison of two proportions using use of self-medication as the exposure variable [19]. We assumed 5% significance, 80% power of the study, proportion of caregivers who had used self-medication was 25.6% versus 74.4% that had not used self-medication. The proportion of caregivers that had used self-medication and had delayed seeking healthcare was estimated at 78.7% while those that did not use self-medication but delayed was 42.1%. Based on these parameters, the estimated sample size was 66. The higher sample size of 384 based on the Kish Leslie formula was considered appropriate for the study.

To explore the barriers and facilitators of prompt care seeking, four Focus Group Discussions (FGDs) each comprising 6 participants were carried out at the end of which saturation

[23] was reached where no new insights were emerging. All eligible study participants agreed to participate when approached.

## Sampling and eligibility criteria

The consecutive method of sampling was used. Eligible caregivers of children under-five with severe pneumonia were enrolled consecutively until the desired sample size was achieved. In line with the WHO definition of severe pneumonia, caregivers whose children, aged 1–59 months, presented with cough and/or difficulty breathing and at least one danger sign of; peripheral arterial oxygen saturation <90%, central cyanosis, severe respiratory distress, inability to drink or breastfeed, vomiting everything, loss of consciousness and convulsion [24] were eligible.

For the FGDs, purposive sampling was used to select caregivers based on their age and ability to use the common local language, Luganda. The choice of age groups for the caregivers was based on previous studies which showed variation in care-seeking between young caregivers below 30 years and those above 30 years [25, 26]. Participants were all female since the majority of the caregivers were female. Stratification by gender was not done because it was not feasible to have male participants in one group as throughout the study only 10 male participants were enrolled.

## Recruitment of study participants

Participants were identified by review of records (patient register) at the paediatric emergency unit. The patient register is managed by the records personnel who sits at the admission desk and is responsible for recording the diagnosis documented in the file by the admitting doctor and assigning file numbers for admitted patients. The personnel updates this register as patients come and are admitted. Once they had been admitted and settled in, the PI or research assistant assessed the identified patients using a screening tool to confirm that they met the WHO case definition of severe pneumonia to verify their eligibility before enrolment.

To explore the barriers and facilitators of prompt healthcare seeking, participants for the FGDs were recruited from caregivers whose children were still undergoing inpatient management for severe pneumonia. These were identified by a review of the inpatient ward registry.

## Operational definitions

**Delay in seeking healthcare for pneumonia.** seeking healthcare at a hospital or health center after 24 hours from when the caregiver recognized the signs and symptoms of pneumonia (cough and/or breathing difficulties).

**Self-medication.** refers to when mothers recognize that their child is ill and give treatment without a medical prescription.

**Severe pneumonia.** cough and/or difficulty in breathing with any one of the following; peripheral arterial oxygen saturation < 90%, central cyanosis, severe respiratory distress, inability to drink or breastfeed or vomiting everything, loss of consciousness, and convulsions [24].

**Caregiver.** someone who attends to the needs of a child and takes responsibility for their well-being. This was an individual who sought care for a sick child.

## Study variables

**Outcome variable.** Delay in seeking healthcare by caregivers was a binary outcome characterized as Delay or No Delay. A child was considered delayed if they were taken to a health

facility more than 24 hours after the of onset pneumonia symptoms; cough and/or difficulty in breathing as reported by the caregiver.

**Independent variables.** *Demographic factors.* The demographic factors considered in this study were; age and sex of the child, age and sex of the caregiver, relationship of caregiver to child, marital status, family size, and birth order of the child.

*The socioeconomic factors.* Education level, occupation, and estimated monthly income of the caregiver.

*Socio-cultural factors and perceptions.* Power relations/decision-making at home, use of traditional medicines, and self-medication.

*Health system factors.* These included distance to the hospital as estimated by the interviewer using google maps after asking for the residence of the caregiver, quality of care, and staff attitudes as perceived by the caregiver.

## Data collection procedures

The identified caregivers were approached to give written informed consent to participate in the study. Quantitative data was collected by the principal investigator and two trained research assistants (nurses) using a pre-tested interviewer administered structured questionnaire. The questionnaire included socio-demographic characteristics of caregivers and children, child clinical characteristics, caregiver health-seeking behavioral characteristics, and perceptions of caregivers on the health system.

Qualitative data was collected by two researchers, one as a facilitator and the other as a note-taker. Focus group discussions were conducted from a quiet side room away from the ward. After obtaining written consent, the moderator used a FGD guide to explore the perceptions of the respondents regarding the barriers and facilitators of seeking prompt healthcare for children under-five with severe pneumonia. The discussions were conducted in Luganda, which is the common local language spoken in the study setting. Each FGD lasted on average 1 hour. The discussions were audio-recorded.

## Data management and analysis

Before data was collected, the case report forms (CRF) were pre-tested on 10 caregivers of children with severe pneumonia in the paediatric emergency unit to check for understanding and completeness and were revised accordingly. Data was collected using hard copy CRF. All completed CRF and other study records such as consent forms and study registers were filed appropriately and locked in the study cabinets to which only the study team had access. Data was checked regularly for quality and completeness by the principal investigator, and any issues arising were addressed immediately or during research team meetings.

Quantitative data was entered in Epidata version 4.2 by two independent data entry clerks and exported to Stata version 14 for cleaning and analysis. Descriptive statistics were used to analyze the participant characteristics and results expressed as proportions, means, and medians, as appropriate. Shapiro Wilks test was used to determine skewness of the different variables and the continuous variables that were not normally distributed were expressed as medians. To determine the prevalence of delay in seeking healthcare, the number of caregivers who brought their children to a health center or hospital beyond 24 hours of onset of symptoms, was divided by the total number of participants, and the result was expressed as a proportion. To determine the factors associated with delay in seeking healthcare, multivariable analysis was undertaken. A logistic regression model was built by including all factors with $p < 0.2$ at bivariate analysis for multivariable analysis. $p \leq 0.05$ at multivariable analysis was considered statistically significant. All statistical tests were 2-sided and considered statistically

significant at α = 0.05. Odds ratios, 95% confidence intervals (CI), and p values are presented. Interaction was assessed using likelihood ratios comparing full and reduced models whereas confounding was assessed by comparing adjusted and unadjusted odds ratios and a predictor variable with a difference of ≥10% was considered as a confounder. Conclusions were drawn based on the adjusted OR with their corresponding 95% confidence intervals. Goodness of fit of the final model was assessed using the Hosmer-Lemeshow test.

The qualitative data was analyzed using content thematic approach in line with the study objectives. Transcripts were read several times to identify themes and sub-themes. Coding was done manually by PE in consultation with JR. This was done by grouping data in line with the themes and sub-themes. Verbatim quotations reflecting participants' views on barriers and facilitators of prompt care seeking for children with severe pneumonia were identified and have been used in the presentation of the study findings.

### Ethical considerations

The study was approved by the Makerere University School of Medicine Research Ethics Committee (REF 2020–192). Administrative clearance was obtained from Mulago National Referral Hospital research ethics committee and the in-charges of the Paediatric wards. Informed consent was obtained from each of the caregivers. The confidentiality of participants was ensured by using serial numbers instead of names and by restriction of access to raw data to the principal investigator and research assistants only. Throughout the conduct of the study, we adhered to the MOH guidelines for the prevention and control of COVID-9.

## Results

### Characteristics of the study participants

A total of 384 caregivers of children under five years of age with severe pneumonia participated in the study. The median age of participants was 27 years (IQR 24–33) with the youngest caregiver being 16 years old and the oldest 75 years. Most caregivers (97.4%) were female. The details are shown in Table 1.

### Characteristics of children less than five years with severe pneumonia

The majority of the children were male (54.4%). The median age was 9 months (IQR 4–18) with the youngest child being one month old and the oldest 57 months old. The caregiver was the mother for most of the children (94%) as shown in Table 2.

### Prevalence of delay in healthcare seeking among caregivers of children less than five years with severe pneumonia at MNRH

The proportion of caregivers who sought care from a hospital or health center, after 24 hours of the onset of pneumonia symptoms was 53.6% (95% CI: 48.6–58.6) as shown in Fig 1 below.

### Clinical characteristics of children

The majority of the children (82%) had severe respiratory distress (severe chest wall-indrawing, grunting, nasal flaring) and 25.5% had hypoxemia ($SpO_2 < 90\%$). Details are shown in Table 3

**Table 1. Socio-demographic characteristics of caregivers of children less than five years with severe pneumonia at Mulago National Referral Hospital (N = 384).**

| Characteristic | Median (IQR) | Frequency(n) | Percentage (%) |
|---|---|---|---|
| **Sex of caregiver** | | | |
| Male | | 10 | 2.6 |
| Female | | 374 | 97.4 |
| **Age of caregiver (years)** | 27(24–33) | | |
| **Education status** | | | |
| None | | 9 | 2.3 |
| Primary | | 142 | 37 |
| Secondary (O and A level) | | 199 | 51.8 |
| University/tertiary | | 34 | 8.9 |
| **Marital status** | | | |
| Single | | 32 | 8.3 |
| Married/cohabiting | | 329 | 85.7 |
| Divorced/separated | | 18 | 4.7 |
| Others (widow/widower) | | 5 | 1.3 |
| **Distance from home to hospital** | | | |
| ≤ 5km | | 138 | 35.9 |
| > 5km | | 246 | 64.1 |
| **Employment status** | | | |
| Employed | | 213 | 55.5 |
| Unemployed | | 171 | 44.5 |
| **Family size** | 5(3–6) | | |
| **Estimated monthly income in Uganda Shillings** | | | |
| > 200,000 (56 USD) | | 81 | 21.1 |
| 100,001–200,000 (28–56 USD) | | 45 | 11.7 |
| 0–100,000 (28 USD) | | 258 | 67.2 |
| **Decision maker at home** | | | |
| Father | | 20 | 5.2 |
| Mother | | 109 | 28.4 |
| Both parents | | 228 | 59.4 |
| Grandparents | | 22 | 5.7 |
| Aunt or uncle | | 4 | 1.0 |
| Others (neighbors) | | 1 | 0.3 |

### Health-seeking behavioral characteristics of caregivers

The majority of the caregivers (82.8%) first sought care from other sources like drug shops, and pharmacies before going to a health center/hospital as shown in Table 4.

### Child factors associated with delay in seeking healthcare

From Table 5, being female and age 2–12 months were associated with delay in seeking healthcare at bivariate analysis.

### Caregiver factors associated with delay in seeking healthcare by caregivers of children under five with severe pneumonia attending MNRH

Living more than 5km from the hospital (AOR 1.94, 95%CI 1.24–3.01, p = 0.003), low-income status (AOR 2.27, 95%CI 1.33–3.86, p = 0.003), and first seeking care from other sources (AOR 3.33, 95%CI 1.85–6.01, p = 0.001) were likely to delay seeking healthcare by caregivers, as shown in Table 6.

**Table 2. Demographic characteristics of children (N = 384).**

| Characteristic | Median (IQR) | Frequency(n) | Percentage (%) |
|---|---|---|---|
| **Sex of the child** | | | |
| Male | | 209 | 54.4 |
| Female | | 175 | 45.6 |
| **Age of child (months)** | 9 (4–18) | | |
| **Child's Birth order** | | | |
| First | | 100 | 26 |
| Second | | 96 | 25 |
| Third | | 73 | 19 |
| Fourth | | 51 | 13.3 |
| Fifth | | 29 | 7.5 |
| Others* | | 35 | 9.1 |
| **Caregiver** | | | |
| Father | | 9 | 2.3 |
| Mother | | 361 | 94.0 |
| Uncle/aunt | | 2 | 0.5 |
| Grandparent | | 11 | 2.9 |
| Other (neighbor) | | 1 | 0.3 |

*Others include above fifth born and unknown. A woman in Uganda has 5 children on average.

## Barriers and facilitators of seeking healthcare promptly for children under five years with severe pneumonia attending MNRH

The facilitators were grouped at the individual level (recognition of symptoms, availability of money for transport), family/relations level (advice from family and friends, supportive spouses), and health organizations level (early identification of danger signs and referral,

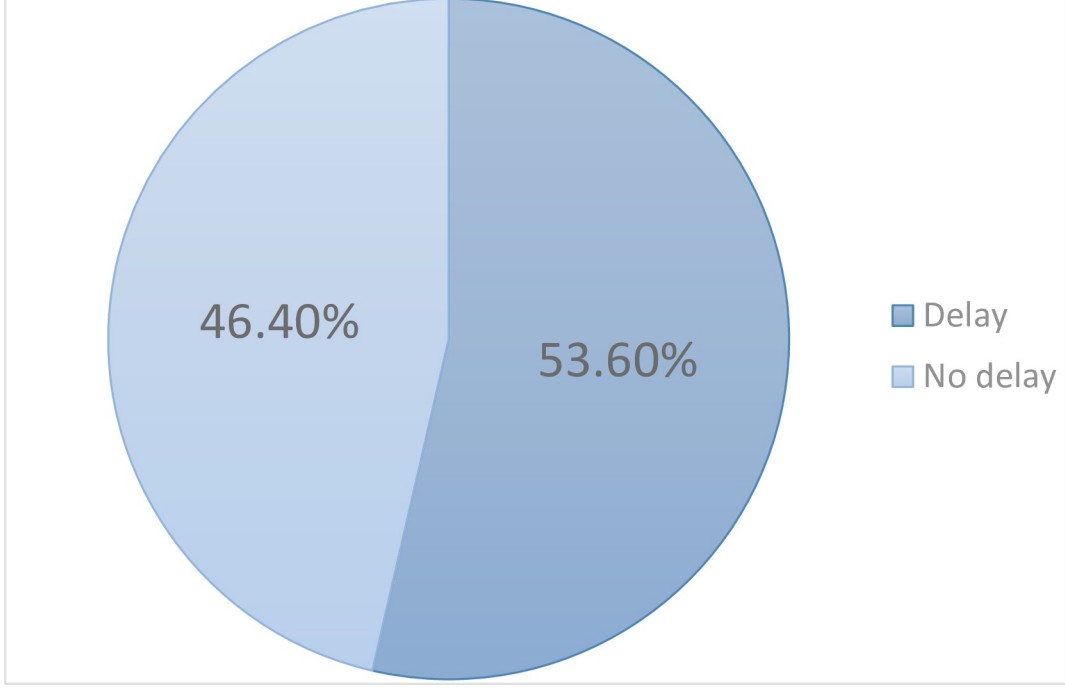

**Fig 1. Prevalence of delay in seeking healthcare by caregivers of children under five with severe pneumonia at MNRH.**

**Table 3. Clinical characteristics of children under five with severe pneumonia at MNRH.**

| Characteristic | Frequency(n) | Percentage (%) |
|---|---|---|
| **Severe respiratory distress** | | |
| No | 69 | 18.0 |
| Yes | 315 | 82.0 |
| **Hypoxemia (SpO$_2$<90%)** | | |
| No | 286 | 74.5 |
| Yes | 98 | 25.5 |
| **Convulsions** | | |
| No | 356 | 92.7 |
| Yes | 28 | 7.3 |
| **Loss of consciousness** | | |
| No | 374 | 97.4 |
| Yes | 10 | 2.6 |
| **Inability to drink/eat or breastfeed** | | |
| No | 234 | 60.9 |
| Yes | 150 | 39.1 |
| **Vomiting everything** | | |
| No | 284 | 74.0 |
| Yes | 100 | 26.0 |
| **Cyanosis** | | |
| No | 383 | 99.7 |
| Yes | 1 | 0.3 |
| **Fever** | | |
| No | 76 | 19.8 |
| Yes | 308 | 80.2 |
| **Others*** | | |
| No | 281 | 73.2 |
| Yes | 103 | 26.8 |

*Others: nasal discharge, nasal congestion

availability of skilled health workers, functional triage systems, good quality of health care provided), as shown in Fig 2.

### Recognition of symptoms of severe illness in the child

Most caregivers were prompted to seek care upon recognition of symptoms of difficulty breathing which they interpreted as indicators of severe illness as some mothers explained:

> "...his breathing was so tight almost like grunting which was initially not there. Now there that is where every parent (myself) realizes that eeh....I need a hospital"-FGD 1, caregivers ≤ 30 years

> "... the child's condition is what forces you to go. The way the child breathes freaks you out" FGD 3, caregivers ≤ 30 years

### Availability of money for transport

Availability of money for transport was another facilitator highlighted by participants. One young mother elaborated:

**Table 4. Health-seeking behavioral characteristics of caregivers of children under five with severe pneumonia at MNRH.**

| Characteristic | Frequency(n) | Percentage (%) |
|---|---|---|
| **Self-medication®** | | |
| No | 316 | 82.3 |
| Yes | 68 | 17.7 |
| **Antipyretics** | | |
| Yes | 167 | 43.5 |
| No | 217 | 56.5 |
| **Antibiotics** | | |
| Yes | 95 | 24.7 |
| No | 289 | 75.3 |
| **Cough syrups** | | |
| Yes | 96 | 30.4 |
| No | 288 | 69.6 |
| **Antimalarial** | | |
| Yes | 52 | 16.5 |
| No | 332 | 83.5 |
| **Use of traditional medicine˚** | | |
| Yes | 92 | 29.2 |
| No | 292 | 70.8 |
| **First form of care sought** | | |
| Elsewhere* | 318 | 82.8 |
| Hospital/ health center | 66 | 17.2 |
| **Had received information about early care seeking#** | | |
| **Yes** | 317 | 82.5 |
| No | 67 | 17.5 |

*Elsewhere: seeking care from drug shops, clinics, pharmacies, spiritual healers, or traditional healers/herbalists.

#Sources of information included health centers, neighbors, media, and relatives.

˚Traditional medicine: included use of local herbs, and food remedies like raw eggs, ground garlic, and ginger to cure disease

®Self-medication: caregivers recognizing that their child is ill and giving treatment without a medical prescription.

> "*If you have the money to start with, you cannot delay. When the child is not okay at night, the mother can't sleep and that is the truth*" FGD 3, caregivers ≤ 30 years

## Advice from relatives and friends

Advice from relatives and friends was another facilitator of prompt care-seeking mentioned by participants. Advice from family and community elders was considered important because they had experience raising and taking care of children and would guide possible courses of action. The family and the community members took the child's health as a part of their responsibility:

> "*there was an elderly woman in our village who told me that most probably my child had pneumonia when I told her about the child's illness. She advised me to take the child to the hospital where there were skilled doctors and even if I do not find medicine; I should go and buy it. So, I went there*"-FGD 1, caregivers ≤ 30 years

**Table 5. Child factors associated with delay in seeking healthcare by caregivers of children less than five years with severe pneumonia attending MNRH.**

| Characteristic | No delay, n (%) n = 178 | Delay, n (%) n = 206 | Unadjusted odds ratio (95% CI) | p-value |
|---|---|---|---|---|
| **Sex of the child** | | | | |
| Male | 110 (52.6) | 99 (47.4) | 1 | |
| Female | 68 (38.9) | 107 (61.1) | 1.75 (1.16–2.62) | **0.007** |
| **Age of child in months** | | | | |
| < 2 | 17 (56.7) | 13 (43.3) | 1 | |
| 2–12 | 84 (41.4) | 119 (58.6) | 1.85 (0.85–4.02) | **0.119** |
| >12–59 | 77 (51.0) | 74 (49.0) | 1.25 (0.57–2.77) | 0.570 |
| **Immunization status** | | | | |
| Up to date* | 170 (47.1) | 191 (52.9) | 1 | |
| Not up to date | 8 (34.8) | 15 (65.2) | 1.67 (0.69–4.04) | 0.255 |
| **Inability to drink/eat or breastfeed** | | | | |
| Yes | 72 (48.0) | 78 (52.0) | 1 | |
| No | 106 (45.3) | 128 (54.7) | 1.12 (0.74–1.68) | 0.605 |
| **Fever** | | | | |
| Yes | 143 (46.4) | 165 (53.6) | 1 | |
| No | 35 (46.1) | 41 (53.9) | 1.01 (0.61–1.68) | 0.953 |
| **Convulsions** | | | | |
| Yes | 10 (35.7) | 18 (64.3) | 1 | |
| No | 168 (47.2) | 188 (52.8) | 0.62 (0.28–1.38) | 0.245 |
| **Vomiting everything** | | | | |
| Yes | 41 (41.0) | 59 (59.0) | 1 | |
| No | 137 (48.2) | 147 (51.8) | 0.75 (0.47–1.18) | 0.213 |

* Up to date immunization status is when a child has received all vaccines appropriate for age according to the Uganda vaccination program schedule.

## Supportive spouses

Participants noted that some fathers actively participated in organizing and arranging for prompt movement to the hospital in cases where they were at home when the child's condition deteriorated. The fathers often accompanied the mother and child to the hospital as one young mother explained:

"*When the child fell ill, I did not see that maybe he was breathing badly, but I woke up and he cried and vomited. His father woke up, dressed up the baby, and said that the next thing we drive the child to hospital*" FGD 1, caregivers ≤ 30 years.

## Prompt identification of danger signs and referral

For some caregivers who first sought care from drug shops or clinics, health workers took a keen interest in identifying the children who needed to be referred which aided in prompt appropriate care-seeking. Those with difficulty in breathing were often quickly referred for further management as one mother explained:

"*When I reached the clinic, the doctor looked at him in the chest, he was breathing fast and there was a depression and he was breathing badly. He said 'I cannot put him here these children are tricky. Let me send you to Mulago'* FDG 3, caregivers ≤ 30 years

**Table 6. Factors associated with delay in healthcare seeking for children under years with severe pneumonia attending MNRH.**

| Characteristic | No delay, n (%) n = 178 | Delay, n (%) n = 206 | Unadjusted odds ratio (95% CI) | p-value | Adjusted odds ratio (95% CI) | p-value |
|---|---|---|---|---|---|---|
| **Age of the caregiver (years)** | | | | | | |
| ≤27 | 84 (42.2) | 115 (57.8) | 1.35 (0.87–2.09) | **0.180** | | |
| 28–37 | 68 (49.6) | 69 (50.4) | 1 | | | |
| 38 and above | 26 (54.2) | 22 (45.8) | 0.83 (0.43–1.61) | 0.589 | | |
| **Distance from hospital^** | | | | | | |
| ≤ 5 km | 80 (58.0) | 58 (42.0) | 1 | | 1 | |
| >5 km | 98 (39.8) | 148 (60.2) | 2.08 (1.36–3.18) | **0.001** | 1.94 (1.24–3.01) | **0.003** |
| **Estimated monthly income in Uganda Shillings©** | | | | | | |
| >200,000 (56 USD) | 51 (63.0) | 30 (37.0) | 1 | | 1 | |
| 100,001–200,000 (28–56 USD) | 16 (20.5) | 62 (79.5) | 3.08 (1.48–6.58) | **0.002** | 3.29 (1.49–7.24) | **0.003** |
| 0–100,000 (28 USD) | 111 (43.0) | 147 (57.0) | 2.25 (1.34–3.76) | **0.002** | 2.27 (1.33–3.86) | **0.003** |
| **Education status** | | | | | | |
| None | 3(33.3) | 6 (66.7) | 3.23 (0.69–15.21) | **0.138** | | |
| Primary | 61 (43.0) | 81 (57.0) | 2.14 (0.99–4.62) | **0.051** | | |
| Secondary (O and A level) | 93 (46.7) | 106 (53.3) | 1.84 (0.87–3.88) | **0.109** | | |
| University/tertiary | 21 (61.8) | 13 (38.2) | 1 | | | |
| **Marital status** | | | | | | |
| Married/cohabiting | 159 (48.3) | 170 (51.7) | 1 | | | |
| Single/widow | 13 (35.1) | 24 (64.9) | 1.73 (0.85–3.51) | **0.131** | | |
| Divorced/separated | 6 (33.3) | 12 (66.7) | 1.87 (0.68–5.10) | 0.221 | | |
| **First form of care sought** | | | | | | |
| Hospital/ health center | 46 (69.7) | 20 (30.3) | 1 | | 1 | |
| Elsewhere* | 132 (41.5) | 186 (58.5) | 3.24 (1.83–5.73) | **0.001** | 3.33(1.85–6.01) | **0.001** |
| **Self-medication** | | | | | | |
| No | 42 (61.8) | 26 (38.2) | 1 | | | |
| Yes | 136 (43.0) | 180 (57.0) | 2.13 (1.25–3.66) | **0.006** | | |
| **Antibiotics use** | | | | | | |
| No | 142 (49.1) | 147 (50.9) | 1 | | | |
| Yes | 36 (37.9) | 59 (62.1) | 1.58 (0.98–2.54) | **0.058** | | |
| **Use of cough syrups** | | | | | | |
| No | 101 (45.9) | 119 (54.1) | 1 | | | |
| Yes | 35 (36.5) | 61 (63.5) | 1.48 (0.90–2.42) | **0.120** | | |

*Elsewhere: drug shops, clinics, pharmacies, spiritual healers, herbalists

^Ministry of Health Uganda recommends a health facility to be within a 5km radius of the population [27].

©Poverty line in Uganda is 1 USD per day (about 100,000 UGX per month) and the World Bank poverty line is 1.9 USD per day (about 200,000 UGX per month) [28].

## Availability of skilled healthcare workers in hospitals

Another facilitator of prompt seeking of healthcare was the availability of skilled healthcare providers at the hospital.

> "*For me to get the qualified doctor to treat my child well, I will not get them from other health facilities. So, I immediately go the government hospital (Mulago) "FGD 1, caregivers ≤ 30 years*

| Facilitators of prompt care seeking for children under five years with severe pneumonia | |
|---|---|
| **Individual**<br><br>• Recognition of symptoms of severe illness in the child<br>• Availability of money for transport | *"…… his breathing was so tight almost like grunting which was initially not there. Now there that is where every parent realizes that eeh… I need a hospital"* |
| **Family /relations**<br><br>• Advice from family, friends and relatives<br>• Supportive spouse | *"there was an elderly woman in our village who told me that most probably my child had pneumonia. She advised me to go to hospital where there were doctors and even if I don't find medicine, I should go and buy it.so I went there"* |
| **Health facility/organizations**<br><br>• Availability of skilled health workers<br>• Functional triage system<br>• Early identification of danger signs and referral<br>• Good quality of healthcare provided | *"for me to get the qualified doctors to treat my child and family well, I will not get them from other health facilities (drug shops, clinics). So, I immediately go to the government hospital (Mulago).."* |

**Fig 2. Facilitators of prompt care-seeking for children under five with severe pneumonia.**

Another young mother further explained:

"*Government hospitals usually have learned (qualified) health workers. They know what they are doing. Even if he doesn't give you medicine but writes on paper, that medicine works*" *FGD 3, caregivers ≤ 30 years*

## Perceived quality of healthcare

Participants reported that the quality of the healthcare they hoped to receive at a hospital/ health facility was integral in encouraging them to seek healthcare promptly. They linked

prompt care-seeking to the past positive experience they had seeking care as one mother explained:

> "*Me I want to thank the health workers here because every time you come, they really care especially when they see that the child is breathing badly. That is why I pass private drug shops and clinics and I decide to come here immediately*" FDG 1, caregivers ≤ 30

## Barriers to prompt care-seeking health care for children under five years with severe pneumonia attending MNRH

Several barriers to prompt healthcare seeking for children with severe pneumonia were mentioned. In line with the three delays model, these were grouped into the first delay, second delay, and third delay. **Delay 1**: Decision to seek help i) poor knowledge of symptoms ii) low level of education iii) self-medication iv) first seeking care at the nearest drug shops and clinics; **Delay 2:** Reach a health facility i) delayed referral ii) long distance to the hospital; **Delay 3**: Receive appropriate help i) unavailability of drugs ii) poor triage iii) unfriendly health workers and iv) unavailability of health workers. As shown in Fig 3.

## Delay 1: Delay in the decision to seek health care services

*Failure to recognize pneumonia symptoms*. Participants described challenges in identifying pneumonia symptoms among their children. Some reported taking a longer period without noticing changes in their child's symptomatology as some mothers explained.

> "*Initially I thought it was just cough. I took him to a clinic and the health worker saw him and said "your child is breathing badly" and I said "that is how he usually is" he said "no this is not normal breathing*" FDG 1, caregivers ≤ 30 years

*Self-medication*. Caregivers mentioned using self-prescribed medications they had previously used to treat respiratory conditions to manage the current pneumonia-like illness. They described using their previous experiences to inform their choices of medicines for their children.

> "*You are used that each time the child gets cough or flu, you normally treat it yourself with Panadol and syrups without first going to the hospital and that may also result in a delay to go to the hospital*" FDG 4, caregivers > 30 years

*Unsupportive spouse*. Participants reported that the lack of participation and support of their male partners delayed their decision to seek care for their children with severe pneumonia. The lack of support and encouragement by their spouses often left the mothers at crossroads as they lacked finances for transport and meeting hospital costs or were told not to go to hospital as some mothers explained:

> "*When your child is sick*, he (father) first stops you from coming to hospital due to fear of asking for money from him. He then tells you to go to buy some tablets and give the child. You are in such a situation yet the child is worsening" FGD 4, caregivers > 30 years

*Poor advice from friends and relatives*. Poor advice by friends and family members was regarded as a cause of delay in taking sick children to the hospital. Caregivers explained that they took advice from relatives, friends, and neighbors because they trusted them and did not

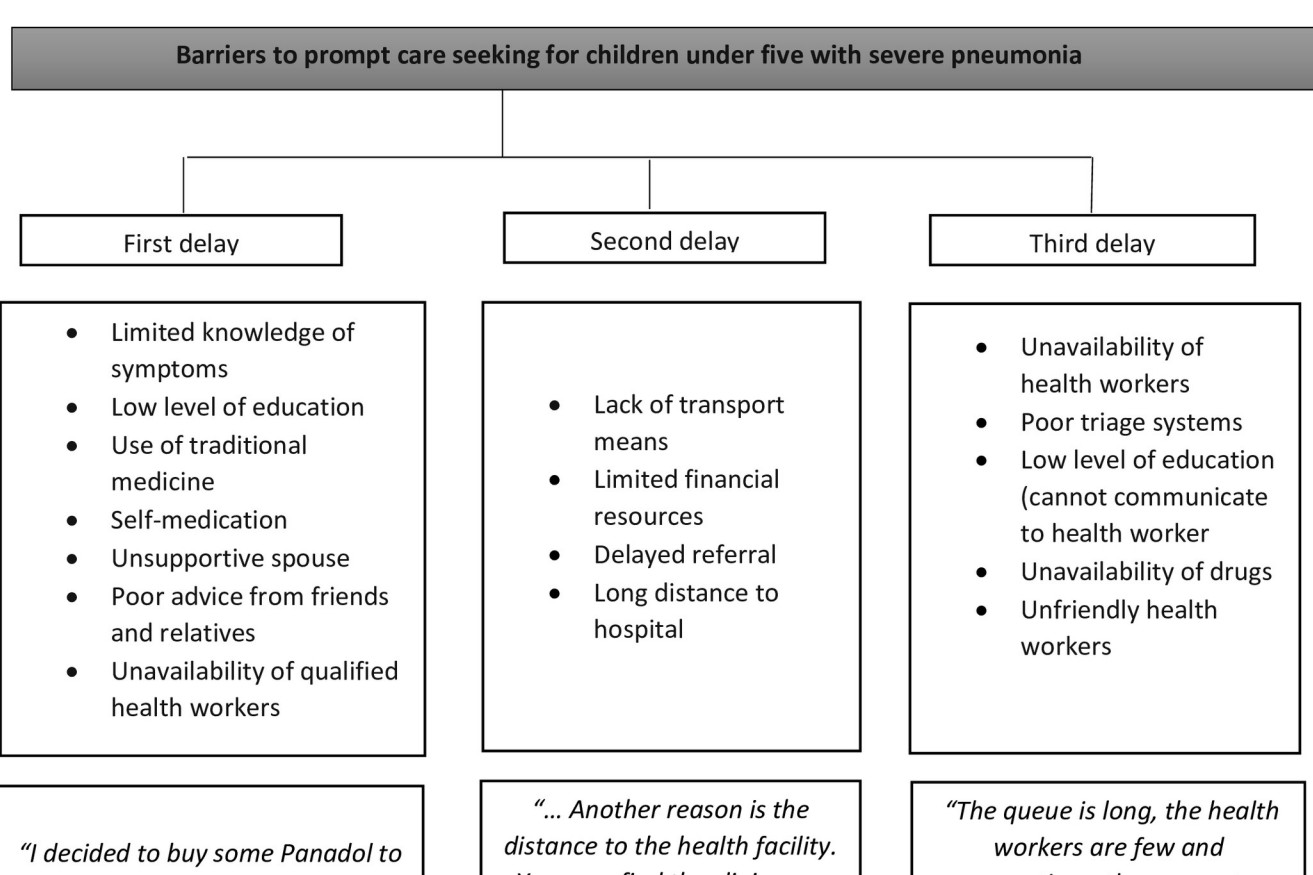

**Fig 3. Barriers to prompt care seeking for children under five with severe pneumonia.**

want to disagree with them. In some cases, the illnesses were perceived as needing herbal medicine thus leading to delays in seeking care for children with severe pneumonia as some mothers explained:

"*A friend can mislead you saying let us go there and pick some herbs that you give to the child. You keep concocting your own medicine while the child is getting problems*"-FGD 3, caregiver ≤ 30 years

"*You may first seek advice from your friends who are mothers and ask for guidance from them about the child's current symptoms before coming to hospital and you delay.*" FDG 2, caregiver > 30 years

*Low level of education.* Another barrier to caregivers deciding to seek healthcare was their low level of education. Instead, caregivers resorted to buying medication such as Paracetamol and syrups from drug shops as some caregivers mentioned:

". . .most of us are illiterate. For me, each time my child would fall sick I would just go and buy some medication from the pharmacy hoping that the child would get better" FDG 4, caregivers > 30 years.

Other barriers to the prompt decision to seek care found in this study were the unavailability of health workers and the fear of high cost of service.

## Second delay: Delay in identifying and reaching a healthcare facility

*Lack of means of transport.* Participants mentioned that they lacked means of transportation to get to a hospital. Many described their ordeal of helplessness when they needed to move especially at night as one mother narrated.

"*Sometimes the transport means prevent us from coming. Because mine started at 10 pm. You cannot move at that time. I had to wait for it to reach morning then I started moving*" FDG 3, caregivers ≤ 30 years.

*Limited financial resources.* Caregivers cited limited access to financial resources as a major barrier to early care-seeking. Mothers reported going to nearby pharmacies to seek quick remedies for their children's ill health. This was because they did not have enough money to cater for transport costs to a health facility.

". . .. I was busy asking for septrin. I thought it could relieve the breathing for some days until I got money for transport to bring me here" FGD 3, caregivers ≤ 30 years

Similarly, some caregivers were unable to reach the hospital due to a lack of money to pay for hospital and transportation. In some cases, donations and fundraising drives from friends and well-wishers were done to enable the caregivers to bring the child to the hospital.

"*We had to first look for money for the hospital before we came. People had to contribute for us to come*"-FGD 4 caregivers > 30 years.

".. You find the child has fallen sick and is stuck at home but all because of the pocket (lack of money)-FGD 1, caregivers ≤ 30 years.

*Delayed referral to a higher-level facility.* Caregivers of children with severe pneumonia experienced delays in referral from drug shops and clinics where they first sought care, to health centers and hospitals. The duration of delay ranged between 3 days to a week before children were referred to Mulago hospital. This (delayed referral) was even experienced in situations where children were not improving on the prescribed medicines. As some mothers narrated:

". . .we spent a week going to a drug shop. The cough and difficulty in breathing were there. The health worker at the drug shop then said "take him to the skilled doctors at Mulago who will work on him because we have given him injections, it has failed, we have given him syrups, they have failed. I have nothing to do for you now, just take the child". When we brought him here, they checked him and said he had pneumonia" FDG 1, caregivers ≤ 30 years.

*Long-distance to hospital.* Participants in the FGDs highlighted challenges of accessibility to the health facility. They often gave up coming to the hospital and preferred to go to the clinics near their home. This is described in the quote:

*"Another reason is the distance to the health facility. You may find that the clinic near home may not give you the desired results yet the better hospital for instance Mulago is far. So, you go to the nearby clinic and end up delaying to reach the hospital."- FGD 4, caregivers > 30 years*

### Third delay: Delay in receiving prompt and adequate health care services

*Low level of education (cannot communicate with health worker).* Some caregivers talked about communication challenges in their interactions with healthcare providers. They mentioned that the language barrier led to an inability to disclose their child's symptoms to the clinician causing delays in the administration of appropriate care as one caregiver mentioned:

*"Now if you find me who did not go to school and you (nurse) tell me in English, I cannot hear and understand it. I do not know it. The nurse asked me questions about my child but I was incapacitated and I started thinking about my child dying from here"-FGD 1, caregivers ≤ 30 years.*

*Poor triage system.* While in the hospital, the clinicians and nursing staff were mentioned as delaying to identify children at high risk of death if not given urgent help. Caregivers with highly critical children spent long waiting times in the queue and in some situations became restless to get the much-needed care. Where the very sick children were triaged, incomplete attention to give them prompt care was often the case distressing the caregivers as two mothers described:

*"My child was very sick. I had waited for so long without being called out of the line (to see the doctor). I just got up at once and entered the doctor's room"–FGD 2, caregivers > 30 years*

*"Some health workers are careless about emergencies. Some health workers are lazy. For example, you may reach the hospital and the health worker may pass by your child thinking it is the usual illness and not bothered. I felt so bad as my child kept worsening in the queue"-FGD 4, caregivers > 30 years*

*Unavailability of drugs.* Participants expressed concern that the health facilities were not well stocked with medicines for the management of pneumonia in children. They cited instances of having to source medications from pharmacies outside the hospital at a cost that led to delays in children receiving appropriate care.

*"Am the mother and father of this child. Sometimes you reach the hospital and there is no medicine and the nurse tells you to go and buy it. However, you have no single coin." FGD 1, caregivers ≤ 30 years*

*Unfriendly health workers.* Another perceived barrier to prompt seeking of health care among the caregivers of children with pneumonia was the bad conduct of some health care providers.

*"The health worker barked at me saying 'Why have you brought the child here without weighing him first?' Even though my child had come in the ambulance on oxygen. I spent much time there as she was tossing me here and here." FGD 4, caregivers > 30 years.*

*Unavailability of health workers.* Sometimes, the caregivers reached the hospital and met few health workers compared to the number of patients in the queue. Their children took

longer than usual to receive diagnoses and the prescribed medication, which left them apprehensive about whether their children will recover from their illnesses on time. As a younger mother told us:

"*The queue is long, the health workers are usually few and sometimes they are not around. I was worried for my child's health the longer I stayed in the line.*"-FGD3, caregivers ≤ 30 years.

## Discussion

This study aimed to determine the prevalence of delay in seeking care for children with severe pneumonia, describe the associated factors, as well as facilitators and barriers. We found that more than half of the caregivers delayed seeking healthcare for their children with severe pneumonia. The prevalence of delay in this study is higher than that found by Kallander and colleagues in a cross-sectional study done in rural eastern Uganda and a study in Rwanda where 47.7% and 35% of the caregivers delayed seeking care for their children with pneumonia symptoms respectively [11, 29]. It is also higher than the prevalence from a similar setting in Kenya in which the prevalence of delayed care seeking was 46.2% [30]. However, other studies in Ethiopia and Kenya found a higher prevalence of more than 60% [17, 31]. The findings from all the above studies clearly show that delay in seeking healthcare for children with pneumonia remains a big problem in sub-Saharan Africa and this leads to deaths that could otherwise be averted by prompt care seeking.

### Factors associated with delay in seeking care

Caregivers who lived more than 5km from the hospital were nearly twice more likely to delay in seeking care compared to those who lived within 5km. Similar findings were noted in studies in Kenya, Peruvian Amazon, and slums in Uganda [16, 17, 32]. WHO recommends that the average distance from homes to the nearest facility should not be more than 5km [27], and this has been shown to improve healthcare seeking [33].

Poverty, as measured by the level of monthly income, was significantly associated with delay in seeking healthcare. Caregivers who earned less than 100,000 Uganda Shillings (28 USD) were 2.27 times more likely to delay seeking care compared to those who earned more. The findings from the qualitative data further confirm that limited financial resources is a key barrier to prompt seeking of healthcare among caregivers of children less than five years with severe pneumonia. These findings are similar to those found in Ethiopia and Nigeria [19, 34]. In low- and middle-income countries like Uganda, poverty influences an individual's choice of when and how to seek healthcare for a sick infant [35]. In many situations, lack of finances leads to the deterioration of children with pneumonia or those suffering from other poverty-related conditions like acute malnutrition and diarrhea causing morbidity and mortality. Our study participants had a low monthly income and no health insurance, which may have predisposed them to delays in seeking care for pneumonia symptoms.

The other significant factor found to be associated with delay in seeking healthcare was first seeking care from other sources like drug shops, pharmacies, and herbalists before going to a hospital. The caregivers who sought care elsewhere before coming to hospital were 3 times more likely to delay compared to those who came straight to hospital. Many reported first going to drug shops, pharmacies, and even herbalists. This finding is consistent with other studies that examined determinants of prompt care-seeking in Africa and Asia [11, 29, 36–39]. Possible explanations for this finding in our study may be due to the high number of drug

shops in this urban area, proximity of drug shops to caregivers' homes, lack of knowledge on adverse consequences of herbal medicine and use of non-prescribed medication as well as perceived benefits of bought medications based on previous experiences of treating similar children [40–43].

In this study, self-medication was not significantly associated with delay in seeking healthcare. However, from the qualitative findings, self-medication was reported as a key barrier to prompt care-seeking. A study by Kallander and colleagues done in Uganda more than 10 years ago found that self-medication as measured by having used antibiotics at home was significantly associated with delay in care-seeking [11]. Similar findings have been reported in studies elsewhere in Africa and South Asia [19, 44, 45]. Self-medication with antibiotics may lead to development of antibiotic resistance [42, 46]. Additionally, self-medication generally may increase the likelihood of children showing no improvements on treatment due to possible unknown interactions between herbal medicines and clinic-prescribed medications.

We found that recognition of symptoms of severe illness, advice from relatives and friends, and early identification of danger signs as facilitators of early care seeking for children with severe pneumonia. According to Bantie and friends, in a cross-sectional study among 356 mothers living in northwest Ethiopia, individuals who were aware of the symptoms of pneumonia and the advantages of seeking care at the hospital early were more likely not to delay [19]. Similar findings were noted in other studies evaluating health seeking behavior determinants [47, 48]. Advice was mainly from fathers of children, friends, and relatives. Consulting fathers and other respected family members like grandparents or in-laws often leads to early care seeking although in some situations may lead to delayed decision making regarding when and where to seek care for some childhood conditions [16, 37, 49, 50].

Availability of skilled health workers was highlighted as a facilitator of prompt care seeking for children with pneumonia. The recommended management of children with pneumonia entails the correct establishment of the severity, appropriate antibiotic use, possible oxygen therapy if indicated, and the ability to change treatment criteria if needed [50, 51]. To achieve this, trained and skilled health workers are pivotal in all levels of care for children with pneumonia.

Relatedly, health worker attitudes may determine an individual's choice to seek healthcare at a given hospital [32]. Rude health workers and poor implementation of triage criteria were barriers to prompt care-seeking in this study. In addition, low levels of health worker staffing and unavailability of medicines led to negative experiences by caregivers while they were at the health facility. Unavailability of medicines and inadequate staffing are common realities in many government health facilities in Uganda and other LMIC areas [16, 49, 50, 52–54], which leads to loss of trust, avoidance of hospital when children are sick and subsequent delays in seeking care. Thus, efforts to improve child health should include improving availability of medicines and ensuring that health facilities have adequate, skilled, and friendly health workers.

Low level of education was noted as a barrier to prompt care-seeking by the study participants. They were either unable to make the decision about early seeking or communicate the child's condition to the health worker. Wambui et al found that highly educated caregivers are more likely to seek immediate healthcare for children with pneumonia symptoms due to possibly their knowledge of symptoms [30]. Similarly, caregivers who had higher levels of education in the Peruvian amazon took their children with pneumonia symptoms to the hospital early [16]. Other studies have also documented related results [19, 55, 56].

Perception of illness entails beliefs about bodily states, psychological and affective reactions, and the strength of perceived representations that inform the behavior of a sick individual [57]. From studies done on delays in seeking care for other childhood illnesses like malaria,

the perception of illness may negatively affect the choice of seeking care [37]. Our study findings revealed that individuals who felt that their child was severely ill sought care quickly compared to those who did not. This may be due to their previous experiences with other children or fear of deterioration of the child at night when they would have been unable to move to the health center. On the contrary, some caregivers reported an inability to identify pneumonia symptoms such as difficulty in breathing, chest in drawing, or abnormal breathing causing delays in seeking care. The caregivers recognized fever and reduced feeding as symptoms their children had which were misleading since other childhood illnesses may present with the same symptoms. These findings re-echo previous findings from studies done in Kenya and Peruvian Amazon where mothers reported that fever and refusal to feed may represent many other childhood illnesses. Thus, it may make it difficult to identify them as symptoms of pneumonia [16, 52].

Our study findings indicated that seeking care from the nearby drug shops or clinics and delays in referral to hospital were also barriers to prompt care seeking. These clinics were near the caregivers despite them having less skilled health workers. Studies done in Uganda have found that caregivers seek care at the clinics for many febrile childhood illnesses including pneumonia [58]. However, delayed referral due to inability to identify poor treatment response may lead to children's deterioration and in the worst cases death [59]. In this study, first seeking care from drug shops or clinics may be explained by the tertiary care status of the study site and a high number of urban drug shops and clinics surrounding the hospital catchment area.

With the majority of mothers seeking care from alternative sources like drug shops, pharmacies, and herbalists before coming to hospital, most children with severe pneumonia continue to receive the much need treatment late.

## Strengths and limitations

This study involved both quantitative and qualitative methods of data collection which offered extensive opportunities for understanding delay in seeking healthcare for children with severe pneumonia. However, findings of this study should be interpreted in light of important limitations. First, it is possible that caregivers might have failed to recall the exact time when the child started experiencing pneumonia symptoms. To counter this, we used precise questions to estimate when symptoms started like asking if symptoms started at about lunch time, night before bed or more than a whole day and night. Secondly, social desirability bias where some caregivers could have said they did not delay for good social ratings thus a likelihood to underestimate the prevalence of delay. However, we ensured privacy during interviews and we were not judgmental which makes us believe the findings. Also, most caregivers in FGDs were free and open to discuss these kinds of things so we believe the impact of social desirability on study results could have been minimal. Lastly, in this study we were unable to explore all health system determinants of delay in receiving appropriate healthcare. In-depth interviews with hospital administrators and health workers would have elucidated perspectives on possible causes of delay in receiving healthcare while in hospital such as poor triage systems and unavailability of medicines.

## Conclusion

More than half of the caregivers delayed seeking healthcare for their children with pneumonia symptoms. As such, many children remain at risk of suffering complications of severe pneumonia including death. Long distance to the hospital, delayed referral, family income, education level of caregiver, recognition of symptoms and their severity as well as family support influence prompt healthcare seeking. Research into effective interventions and health system

strengthening to improve prompt healthcare seeking for children with severe pneumonia is highly needed in Uganda and other low-income settings that may have similar challenges

## Acknowledgments

We acknowledge the great contribution of the caregivers who participated in this study. We thank the research assistants who collected the data used in this publication.

## Author Contributions

**Conceptualization:** Phiona Ekyaruhanga, Rebecca Nantanda, Hellen T. Aanyu, Joseph Rujumba.

**Data curation:** Phiona Ekyaruhanga, Mukeere John, Palma Aceng.

**Formal analysis:** Phiona Ekyaruhanga, John Mukisa, Judith Amutuhaire Ssemasaazi, Joseph Rujumba.

**Funding acquisition:** Phiona Ekyaruhanga.

**Investigation:** Phiona Ekyaruhanga.

**Methodology:** Phiona Ekyaruhanga, Rebecca Nantanda, Hellen T. Aanyu, John Mukisa, Judith Amutuhaire Ssemasaazi, Joseph Rujumba.

**Project administration:** Phiona Ekyaruhanga.

**Resources:** Phiona Ekyaruhanga.

**Software:** John Mukisa.

**Supervision:** Rebecca Nantanda, Hellen T. Aanyu, Joseph Rujumba.

**Validation:** Phiona Ekyaruhanga, John Mukisa.

**Writing – original draft:** Phiona Ekyaruhanga.

**Writing – review & editing:** Rebecca Nantanda, Hellen T. Aanyu, Mukeere John, Palma Aceng, Joseph Rujumba.

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
