## [Decision Letter · Decision Letter 0]

7 Feb 2023

PONE-D-22-33296Delay in healthcare seeking for children with severe pneumonia at Mulago National Referral Hospital: a mixed methods cross-sectional studyPLOS ONE

Dear Dr. Phiona Ekyaruhanga,

Thank you for submitting your manuscript to PLOS ONE. After careful consideration, we feel that it has merit but does not fully meet PLOS ONE’s publication criteria as it currently stands. Therefore, we invite you to submit a revised version of the manuscript that addresses the points raised during the review process.

ACADEMIC EDITOR: Please go thoroughly and address the comments of reviewers. The manuscript is important but need to solve the technical and scientific issues raised by reviewers. 

We look forward to receiving your revised manuscript.

Kind regards,

Kshitij Karki, MPH, MA

Academic Editor

PLOS ONE

Journal Requirements:

Additional Editor Comments:

Thank you for your important research. Please go through the suggestions of the reviewers and revise the manuscript as suggested.

Reviewers' comments:

Reviewer's Responses to Questions

**Comments to the Author**

1. Is the manuscript technically sound, and do the data support the conclusions?

Reviewer #1: Yes

Reviewer #2: No

2. Has the statistical analysis been performed appropriately and rigorously? 

Reviewer #1: Yes

Reviewer #2: No

3. Have the authors made all data underlying the findings in their manuscript fully available?

Reviewer #1: Yes

Reviewer #2: No

4. Is the manuscript presented in an intelligible fashion and written in standard English?

Reviewer #1: Yes

Reviewer #2: No

5. Review Comments to the Author

Reviewer #1: Revise the sampling procedure, it say convenient method for the quantitative. For the qualitative they used four FGDs why four? In the analysis part, it is better to run both Bivariable and multivariable.

Reviewer #2: In its current form, this manuscript is not publishable. More specific:

1. Introduction is not specific and very long

2. Methods are not described appropriately, including how sample size was calculated

3. Analysis was not performed appropriately. More appropriate to conduct multilevel modelling

4. Discussion section is not very well written

6. PLOS authors have the option to publish the peer review history of their article (what does this mean?). If published, this will include your full peer review and any attached files.

Reviewer #1: No

Reviewer #2: No

---

## [Author Response · Author response to Decision Letter 0]

12 May 2023

May 10, 2023

Academic Editor,

PLOS ONE

Dear Editor,

Re: Response to reviewers’ comments on our manuscript titled: Delay in healthcare seeking for young children with severe pneumonia at Mulago National Referral Hospital, Uganda: a mixed methods cross-sectional study. (Ref No: PONE-D-22-33296)

We are grateful for the comments and suggestions on our manuscript by the reviewers. We have addressed them and specified our response to each of the reviewers’ comments.

Reviewer #1: 

Comment: Revise the sampling procedure, it say convenient method for the quantitative. 

Response: We thank the reviewer for highlighting this correction. The sampling method has been revised to state that the consecutive method of sampling was used instead of convenience method (See page 4, line 129). The children were then enrolled consecutively until the desired sample size was realized. Studies on sampling methods have indicated that consecutive sampling is feasible and offers representative samples in clinical studies (1). So, the assumption was that the children that were sampled were representative of those that are seen at the emergency paediatric unit of Mulago hospital.

Comment: For the qualitative they used four FGDs why four? 

Response: Four FGDs were carried out at the end of which saturation (2) was reached where no new insights were emerging. This clarification has been made (See page 4, lines 124 – 126). 

Comment: In the analysis part, it is better to run both Bivariable and multivariable

Response: The analysis section has been elaborated extensively and tables updated to describe the multilevel modeling done detailing both bivariable and multivariable analysis. A logistic regression model was built by including all factors with p<0.2 at bivariable analysis for multivariable analysis. p≤0.05 at multivariable analysis was considered statistically significant. All statistical tests were 2-sided and considered statistically significant at α = 0.05. Odds ratios, 95% confidence intervals (CI) and p values are presented. Interaction was assessed using likelihood ratios comparing full and reduced models whereas confounding was assessed by comparing adjusted and unadjusted odds ratios and a predictor variable with a difference of ≥10% was considered as a confounder. Conclusions were drawn based on the adjusted OR with their corresponding 95% confidence intervals. Goodness of fit of the final model was assessed using the Hosmer-Lemeshow test. (See page 7, lines 187 – 202, table 5 and table 6)

Reviewer #2: In its current form, this manuscript is not publishable. More specific:

Comment: 1. Introduction is not specific and very long

Response: We thank the reviewer for highlighting this concern. The introduction section has extensively been revised, shortened, and made more focused. It has been re-written to focus on the burden of pneumonia, delay in seeking care and associated factors. In the initial version of the manuscript, it was 2.5 pages and has now been reduced to 1.5 pages. (See pages i- ii, lines 55-93)

Comment: 2. Methods are not described appropriately, including how sample size was calculated

Response: Thank you for this comment. We have revised the methods section entirely and also made explicit how the sample size was calculated. The sample size to determine the prevalence of delay in seeking healthcare was estimated using the formula for single proportions. We assumed 95% level of confidence, 5% sampling error, proportion of caregivers that delayed seeking healthcare for their children of 47.7% (3). This gave an estimated sample size of 384. In addition, we estimated the sample size for determining factors associated with delay in seeking healthcare based on the formular for comparison of two proportions using use of self-medication as the exposure variable (4). We assumed 5% significance, 80% power of the study, proportion of caregivers who had used self-medication was 25.6% versus 74.4% that had not used self-medication. The proportion of caregivers that had used self-medication and had delayed seeking healthcare was estimated at 78.7% while those that did not use self-medication but delayed was 42.1%. Based on these parameters, the estimated sample size was 66. The higher sample size of 384 based on the modified Kish Leslie formula was considered appropriate for the study (See page 3 - 4, lines 111-127). 

Comment: 3. Analysis was not performed appropriately. More appropriate to conduct multilevel modelling

Response: More details on analysis have been added in the revised manuscript. Specifically, details on how multilevel modeling was done have been made explicit and tables updated (tables 5 and 6). 

Descriptive statistics were used to analyse the participant characteristics and results expressed as proportions, means and medians, as appropriate. Shapiro wilks test was used to determine skewness of the different variables and the continuous variables that were not normally distributed were expressed as medians. To determine the prevalence of delay in seeking healthcare, the number of caregivers who took their children to a health center or hospital beyond 24 hours of onset of symptoms, was divided by the total number of participants and the result was expressed as a proportion. To determine the factors associated with delay in seeking healthcare, multivariable analysis was undertaken. A logistic regression model was built by including all factors with p<0.2 at bivariate analysis for multivariable analysis. P≤0.05 at multivariable analysis was considered statistically significant. All statistical tests were 2-sided and considered statistically significant at α = 0.05. Odds ratios, 95% confidence intervals (CI) and p values are presented. Interaction was assessed using likelihood ratios comparing full and reduced models whereas confounding was assessed by comparing adjusted and unadjusted odds ratios and a predictor variable with a difference of ≥10% was considered as a confounder. Conclusions were drawn based on the adjusted OR with their corresponding 95% confidence intervals. Goodness of fit of the final model was assessed using the Hosmer-Lemeshow test.

The qualitative data was analysed using content thematic approach in line with the study objectives. Transcripts were read several times to identify themes and sub-themes. Coding was done manually by PE in consultation with JR. This was done by grouping data in line with the themes and sub-themes. Verbatim quotations reflecting participants’ views on barriers and facilitators for prompt care seeking for children with severe pneumonia were identified and have been used in the presentation of the study findings. (See pages 7- 8, lines 204 - 226)

Comment: 4. Discussion section is not very well written

Response: We thank the reviewer for this comment. We have revised and strengthened the discussion section highlighting the key results, what they mean and compared with other studies. (See pages 29 - 34)

1. Bjørn M, Brendstrup C, Karlsen S, Carlsen JE. Consecutive screening and enrollment in clinical trials: the way to representative patient samples? Journal of cardiac failure. 1998;4(3):225-30.

2. Hennink MM, Kaiser BN, Marconi VC. Code saturation versus meaning saturation: how many interviews are enough? Qualitative health research. 2017;27(4):591-608.

3. Källander K, Hildenwall H, Waiswa P, Galiwango E, Peterson S, Pariyo G. Delayed care seeking for fatal pneumonia in children aged under five years in Uganda: a case-series study. Bulletin of the World Health Organization. 2008;86:332-8.

4. Bantie GM, Meseret Z, Bedimo M, Bitew A. The prevalence and root causes of delay in seeking healthcare among mothers of under five children with pneumonia in hospitals of Bahir Dar city, North West Ethiopia. BMC pediatrics. 2019;19(1):482.

---

## [Editor Report · Decision Letter 1]

11 Jul 2023

PONE-D-22-33296R1Delay in healthcare seeking for young children with severe pneumonia at Mulago National Referral Hospital, Uganda: a mixed methods cross-sectional study.PLOS ONE

Dear Dr. Ekyaruhanga,

Thank you for submitting your manuscript to PLOS ONE. After careful consideration, we feel that it has merit but does not fully meet PLOS ONE’s publication criteria as it currently stands. Therefore, we invite you to submit a revised version of the manuscript that addresses the points raised during the review process. Please submit your revised manuscript by Aug 25 2023 11:59PM. If you will need more time than this to complete your revisions, please reply to this message or contact the journal office at plosone@plos.org. Please include the following items when submitting your revised manuscript:A rebuttal letter that responds to each point raised by the academic editor and reviewer(s). You should upload this letter as a separate file labeled 'Response to Reviewers'.A marked-up copy of your manuscript that highlights changes made to the original version. You should upload this as a separate file labeled 'Revised Manuscript with Track Changes'.An unmarked version of your revised paper without tracked changes. You should upload this as a separate file labeled 'Manuscript'.If applicable, we recommend that you deposit your laboratory protocols in protocols.io to enhance the reproducibility of your results. Protocols.io assigns your protocol its own identifier (DOI) so that it can be cited independently in the future. For instructions see: https://journals.plos.org/plosone/s/submission-guidelines#loc-laboratory-protocols. Additionally, PLOS ONE offers an option for publishing peer-reviewed Lab Protocol articles, which describe protocols hosted on protocols.io. Read more information on sharing protocols at https://plos.org/protocols?utm_medium=editorial-email&utm_source=authorletters&utm_campaign=protocols.

We look forward to receiving your revised manuscript.

Kind regards,

Kshitij Karki, MPH, MA

Academic Editor

PLOS ONE

Journal Requirements:

Additional Editor Comments :

Please check the spelling and grammatical errors. Also, revise the tables with row percentage (For example: if you are comparing delay and not delay with other variables). Thank you

---

## [Author Response · Author response to Decision Letter 1]

21 Aug 2023

August 8, 2023

Academic Editor,

PLOS ONE

Dear Editor,

Re: Response to the editors’ comments on our manuscript titled: Delay in healthcare seeking for young children with severe pneumonia at Mulago National Referral Hospital, Uganda: a mixed methods cross-sectional Study. (Ref No: PONE-D-22-33296R1)

We appreciate the comments on our manuscript. We have addressed them and specified our response to each of the comments.

Comment: Please review your reference list to ensure that it is complete and correct. If you have cited papers that have been retracted, please include the rationale for doing so in the manuscript text, or remove these references and replace them with relevant current references. Any changes to the reference list should be mentioned in the rebuttal letter that accompanies your revised manuscript. If you need to cite a retracted article, indicate the article’s retracted status in the References list and also include a citation and full reference for the retraction notice.

Response: We thank the reviewer for highlighting this correction. We have reviewed the reference list and corrected any misspelled words. We are sure the list is complete and correct. No cited papers have been retracted.

Comment: Please check the spelling and grammatical errors. 

Response: We have read the whole manuscript and corrected all the spelling and grammatical errors as marked up in the track changes. 

Comment: Also, revise the tables with row percentage (For example: if you are comparing delay and not delay with other variables). Thank you.

Response: The tables with row percentages have been revised to have percentages of caregivers with a variable who delayed versus those with the same variable who did not delay. (See page 15 and 17, table 5 and table 6)

---

## [Editor Report · Decision Letter 2]

29 Aug 2023

Delay in healthcare seeking for young children with severe pneumonia at Mulago National Referral Hospital, Uganda: a mixed methods cross-sectional study.

PONE-D-22-33296R2

Dear Dr. Phiona Ekyaruhanga,

We’re pleased to inform you that your manuscript has been judged scientifically suitable for publication and will be formally accepted for publication once it meets all outstanding technical requirements.

Kind regards,

Kshitij Karki, MPH, MA

Academic Editor

PLOS ONE

Additional Editor Comments (optional):

Please use consistent font in text and table as per guideline. Also, check and review the interpretation in results based on the revised tables.
---

## [Editor Report · Acceptance letter]

2 Oct 2023

PONE-D-22-33296R2 

Delay in healthcare seeking for young children with severe pneumonia at Mulago National Referral Hospital, Uganda: a mixed methods cross-sectional study. 

Dear Dr. Ekyaruhanga:

I'm pleased to inform you that your manuscript has been deemed suitable for publication in PLOS ONE. Congratulations! Your manuscript is now with our production department. 

Kind regards, 

on behalf of

Dr. Kshitij Karki 

Academic Editor

PLOS ONE